# Geometry-Aware Depth-Guided Explainable Multimodal Polyp Size Estimation: A Fusion Model Beyond RGB

**Krispian Lawrence**[1]                                                        KRISPIAN@MIT.EDU

**Usha Goparaju**[1]                                   USHA.GOPARAJU@SCREENWITHPEEK.COM

**Luis Lamb**[2]                                                        LUISLAMB@ALUM.MIT.EDU

[1] *Equitable Technologies, Cambridge, MA, USA*

[2] *Catholic Institute of Technology, Cambridge, MA, USA*

**Editors:** Accepted for publication at MIDL 2026

## Abstract

Accurately estimating the physical size of colorectal polyps from monocular endoscopy is difficult due to scale ambiguity, viewpoint distortions, and strong inter-patient variability. We introduce MPSE, a geometry-aware, depth-guided multimodal framework that jointly leverages RGB appearance, monocular depth cues, and interpretable geometry descriptors to produce reliable and clinically calibrated size estimates. Central to MPSE is a geometry-as-query fusion block that selectively attends to depth and RGB features, and a Scale Consistency Block (SCB) that models agreement between 2D footprint–derived and 3D depth–derived cues, reducing size bias under severe distribution imbalance. The model is trained with a primary regression objective supported by an auxiliary threshold-based classification loss that stabilizes predictions near clinically important cutoffs. On our clinical dataset, MPSE achieves a mean absolute error of 0.93 mm and a polyp-level F1 score of 0.87 at the clinically critical 5 mm threshold, demonstrating accurate and clinically reliable size estimation in endoscopy.

**Keywords:** polyp size estimation, depth fusion, geometry-awareness, multimodal learning, endoscopy

## 1. Introduction

Colonoscopy is central to preventing colorectal cancer (CRC), which remains a major global health burden with nearly two million new cases and high mortality each year (Sung et al., 2021). The early detection and removal of precancerous polyps substantially reduce CRC incidence and death rates (Nusko et al., 2000; Intissar and Yassine, 2025). Among polyp characteristics assessed during endoscopy, *polyp size* is one of the strongest predictors of malignant potential. Large-scale investigations, such as that of (Nusko et al., 2000) show that polyps ≤5 mm have negligible malignant risk, whereas lesions 26–35 mm are malignant in 42.4% of cases and those larger than 35 mm reach malignancy rates of 75% . Consequently, contemporary guidelines (e.g., ESGE and USMSTF ) rely on size thresholds at 5 mm and 10 mm to determine resection technique and surveillance intervals (Ferlitsch et al., 2024; Gupta et al., 2020). Accurate size estimation is also essential for implementing resect-and-discard and diagnose-and-leave strategies (Jeon and Kim, 2025). However, conventional visual estimation during colonoscopy suffers from substantial interobserver variability and lack of standardization, leading to frequent overestimation or underestimation of lesion size (Song et al., 2025). Endoscopic magnification, optical distortion, inconsistent use of

reference tools, shrinkage of resected tissue, and piecemeal excision further compound these inaccuracies.

Recent advances in image-based measurement systems such as Virtual Scale Endoscopy (VSE) and laser-based virtual rulers have demonstrated improved reproducibility compared to traditional visual estimation (Jeon and Kim, 2025). Yet, these technologies require specialized hardware and are not widely available across clinical settings. Parallel progress in artificial intelligence (AI) has transformed endoscopic detection, classification, and segmentation (Dhali et al., 2025; Kao et al., 2025), but polyp *size estimation* remains comparatively underexplored. Estimating 3D dimensions from a single 2D image is an inherently ill-posed problem, polyps with vastly different true diameters may appear visually similar depending on camera–polyp distance, as showed in recent depth-based studies (Hwang et al., 2021; Liu et al., 2025; Yang et al., 2025). Emerging datasets such as Polyp-Size (Song et al., 2025) highlight the magnitude of measurement inconsistencies even among experts, with mean relative errors reaching up to 77% in clinical practice (Atalaia-Martins et al., 2019). These limitations shows the need for robust, reproducible, and clinically aligned AI-driven methods for accurate polyp size estimation using standard endoscopic video. This paper is organized as follows. Section 2 reviews related work. Section 3 presents the proposed methodology. Section 4 describes the dataset and experimental setup. Section 5 reports and discusses the results. Section 6 concludes the study and outlines future directions.

## 2. Related Work

Recent advances in computational endoscopy have significantly advanced tasks such as polyp *detection*, *segmentation*, and *histology prediction*, largely driven by deep learning. However, *polyp size estimation* despite its critical role in risk stratification and surveillance interval decisions remains underexplored. Among the relatively few studies that address size estimation, a major unresolved challenge lies in the definition of "ground truth." Reported size labels are typically derived from visually estimated measurements by endoscopists (Wang et al., 2024a), scale referencing using biopsy forceps or virtual tools (Shimoda et al., 2022a), post-resection pathology (which suffers from shrinkage and deformation), or 3D measurements from CT colonography (Yee et al., 2001). Each of these sources introduces systematic uncertainty due to scale ambiguity, tissue deformation, or lack of direct metric correspondence creating inherent noise in both training labels and evaluation baselines.

Traditional approaches to size estimation have relied on tactile references or vision-based overlays. Early systems such as Virtual Scale Endoscopy (VSE) incorporated calibrated forceps or laser rulers to infer polyp size during colonoscopy (Nakatani et al., 2007; Yoshioka et al., 2021), offering improved accuracy over visual estimation but depending heavily on ideal pose, fixed distances, and planar assumptions. Shimoda et al. (Shimoda et al., 2022b) and Djinbachian et al. (Djinbachian et al., 2023) demonstrated that VSE can reduce misclassification at clinical thresholds (e.g., 5 or 10 mm), but these systems still struggle with morphologically irregular polyps or distorted views. More recent evaluations (Minakata et al., 2024) confirm reduced inter-observer variability using such tools, but highlight limitations in image-only processing and lack of structural priors. Critically, none of these works incorporate cross-frame temporal consistency, multi-view geometry, or multimodal integration of tool and scene context factors essential for robust measurement in dynamic

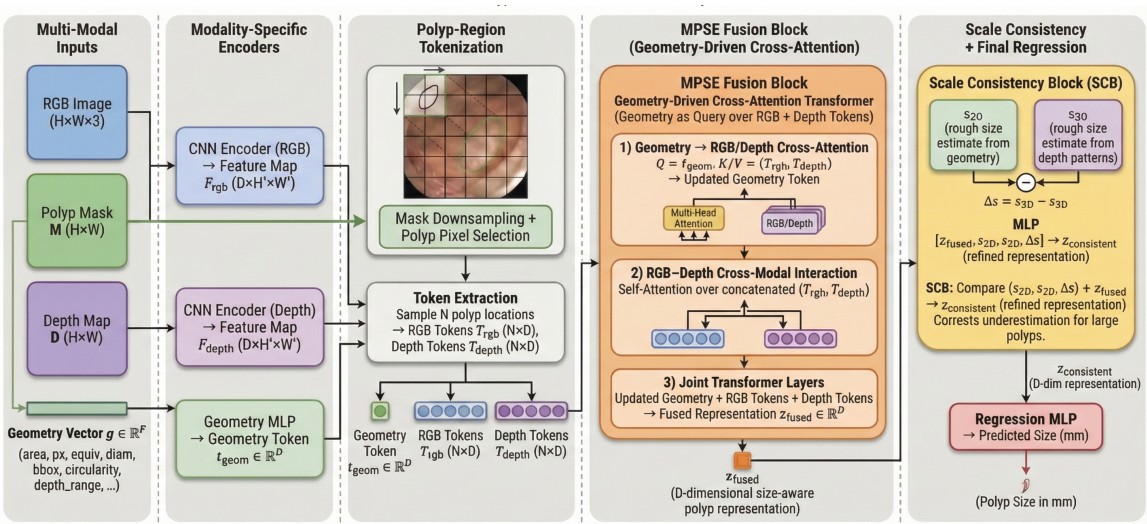

Figure 1: Overview of MPSE. RGB, depth, and analytic geometry descriptors are fused via a geometry-as-query attention block, followed by a Scale Consistency Block (SCB).

or anatomically complex scenes. These limitations motivate the shift toward learning-based approaches that can encode semantic, geometric, and spatial features jointly.

Deep learning-based methods have recently begun addressing size estimation, both in colorectal and esophageal endoscopy. Abdelrahim et al.Du et al. (Du et al., 2024) leveraged monocular metric depth estimation and 3D reconstruction to infer polyp size, demonstrating the value of depth-aware representations but without explicit lesion geometry modeling or cross-cue consistency. Ruano et al. (Ruano et al., 2024) used a shape-from-shading model to estimate polyp size from a single image, but relied on restrictive illumination assumptions and lacked temporal or multimodal integration. (Abdelrahim et al., 2022) combined structure-from-motion with CNNs to classify polyps into binary size categories, achieving 85.2% accuracy outperforming expert endoscopists but relied on small, constrained video datasets. Kwak et al. (Kwak et al., 2022) introduced a W-Net regression model showing strong agreement with size labels (CCC = 0.961), yet their method lacked depth inference, temporal reasoning, or uncertainty quantification. More clinically oriented systems such as ENDOANGEL-CPS (Wang et al., 2024b) demonstrated real-time performance with 89.9% accuracy and reduced inappropriate surveillance recommendations, but functioned as black-box predictors without leveraging procedural or spatial context. In parallel, esophageal variceal studies (Fang et al., 2024; Mao et al., 2024) used VR-assisted or AI-guided measurements to reduce evaluation time and improve outcome prediction, but similarly failed to account for shape deformation, pressure-induced size changes, or vessel morphology factors critical for pathophysiological relevance. Despite these advances, no existing system explicitly fuses RGB visual appearance, segmentation-derived geometry, and depth or spatial priors within a unified, trainable architecture. Our work addresses this gap by *proposing a multimodal fusion framework that integrates temporal visual features, geometric cues, and tool-aware context to deliver anatomically grounded and metrically meaningful polyp size estimation.*

## 3. Method: MPSE - Multimodal Polyp Size Estimation

Estimating the physical size of a colorectal polyp from a single endoscopic frame is intrinsically ill-posed: supervision is weak a single scalar diameter in millimeters while the cues that govern size perception are structurally heterogeneous. Accurate estimation simultaneously depends on the lesion's two-dimensional footprint, its three-dimensional protrusion and local curvature, and the photometric and viewpoint factors that shape its appearance. Crucially, no single modality is sufficient on its own. MPSE coordinates these complementary but inconsistent signals through a unified representation. To this end, MPSE derives analytic morphology from the segmentation mask, extracts dense RGB and monocular-depth evidence over the lesion surface, and fuses them through a **geometry-driven transformer** in which an explicit geometry token guides cross-modal attention. A subsequent **Scale Consistency Block (SCB)** reconciles 2D and 3D size cues, correcting the systematic underestimation of large lesions observed in clinical practice and prior learning-based approaches.

### 3.1. Multimodal Cue Extraction

Each frame provides an RGB image $I$, from which a pretrained segmentation network like Polyp-PVT (Dong et al., 2023) extracts a binary mask $M$ delineating the polyp. Beyond defining spatial support, $M$ enables computation of a compact set of analytic geometry descriptors $g$ that summarise the lesion's 2D morphology. We obtain (i) pixel area and equivalent diameter, (ii) axis-aligned aspect ratios capturing elongation, (iii) circularity from contour moments, and (iv) a Laplacian-based boundary-sharpness score. These descriptors arise from deterministic operations (connected components, contour geometry, and moment estimation), yielding invariance to rotation, mild affine distortions, and moderate illumination changes. Crucially, they are not heuristic handcrafted features: they encode the same structural cues endoscopists use when visually estimating size and remain stable under appearance degradations such as blur, specularities, and mucosal texture.

However, as showed in Fig. 1, two lesions with nearly identical 2D footprints may differ substantially in true physical size due to changes in protrusion height or camera–lesion distance, information not recoverable from $M$ alone. To supply these missing 3D cues, we estimate a pseudo-metric depth map $D$ using EndoDAC (Cui et al., 2024), an endoscopy-specific monocular depth model whose photogeometric priors better capture relative curvature, protrusion, and coarse viewing distance compared with generic depth networks.

From the RGB frame $I$ and depth map $D$, modality-specific encoders $F_{\text{rgb}}$ and $F_{\text{depth}}$ (both ResNet–18 backbones) extract dense feature maps, which are projected into $D$-dimensional tokens through linear projections. The RGB and depth token sets are thus defined compactly as

$$T_{\text{rgb}} = \Pi_{\text{rgb}}(\text{flatten}(F_{\text{rgb}}(I))), \qquad T_{\text{depth}} = \Pi_{\text{depth}}(\text{flatten}(F_{\text{depth}}(D))),$$

yielding $T_{\text{rgb}}, T_{\text{depth}} \in \mathbb{R}^{N \times D}$ with $N = H'W'$.

The analytic geometry vector $g \in \mathbb{R}^F$ (area, equivalent diameter, circularity, aspect ratios, boundary sharpness) is mapped into the same $D$-dimensional space using a two-layer MLP,

$$t_{\text{geom}} = W_2\, \sigma(W_1 g + b_1) + b_2, \tag{1}$$

producing a single geometry token aligned with the RGB and depth embeddings.

## 3.2. Polyp-Region Tokenization

Colonoscopy frames contain extensive background mucosa whose appearance varies widely across patients and illumination conditions but contributes no meaningful information for estimating physical size. Allowing these regions to generate transformer tokens would inflate the attention space, introduce modality-incoherent noise, and weaken optimization dynamics. MPSE therefore tokenizes only the *polyp surface*. The segmentation mask $M$ is downsampled to match the backbone resolution of the RGB and depth encoders, producing aligned feature maps $F_{\text{rgb}}$ and $F_{\text{depth}}$. From the foreground coordinates, we uniformly sample $N$ positions that capture the lesion footprint. At each selected location $(u, v)$, RGB and depth descriptors are extracted from the corresponding feature tensors and projected into embedding space, yielding synchronized token sets.

## 3.3. MPSE Fusion Block

The central hypothesis behind MPSE is that *global geometry should guide multimodal fusion.* Geometry provides footprint shape, scale priors, and coarse depth statistics, whereas RGB and depth tokens provide dense but highly local evidence. Treating modalities symmetrically forces the model to infer global structure from local cues alone, a setting that empirically produces unstable fusion and biased size estimates. MPSE therefore assigns the geometry token a privileged role: it is the *sole query* in the first attention stage, enabling explicitly directed global-to-local reasoning.

**(1) Geometry-as-Query Attention.** The geometry token attends over all RGB and depth tokens to extract the local evidence most consistent with the global morphological prior:

$$\tilde{t}_{\text{geom}} = \text{MHAttn}\big(Q = t_{\text{geom}}, \ K, V = [T_{\text{rgb}}, T_{\text{depth}}]\big) \tag{2}$$

This design substantially stabilizes optimization and reduces fusion ambiguity compared to RGB- or depth-driven query streams.

**(2) RGB–Depth Cross-Modal Interaction.** RGB and depth tokens are then refined through a shared multi-head self-attention layer, allowing shading-based appearance cues and curvature-based depth cues to mutually correct and reinforce one another. This bidirectional alignment mitigates cases where one modality is noisy or visually ambiguous.

**(3) Joint Transformer Encoding.** Finally, the updated geometry token is prepended to the refined RGB/depth tokens and passed through $L = 2$ lightweight TransformerEncoder layers. The output token at position 0 corresponding to the geometry slot acts as the fused representation $z_{\text{fused}}$, summarizing appearance, morphology, and 3D topology in a geometry-aware manner.

This three-stage pipeline yields a stable and interpretable multimodal embedding: geometry determines *where* to attend, RGB and depth reconcile local inconsistencies, and the joint encoder produces a coherent representation suitable for scale-consistent regression.

### 3.4. Scale Consistency Block

Even with strong multimodal fusion, 2D footprint cues and 3D depth cues may disagree, particularly for flat or large lesions or under oblique viewing angles. These conflicts drive the network toward mid-range predictions and cause systematic underestimation in the clinically important $> 5\,\text{mm}$ and $> 10\,\text{mm}$ ranges. The Scale Consistency Block (SCB) makes this disagreement explicit by computing two internal size proxies. A *2D footprint estimate $s_{2D}$* is obtained by applying a small MLP to the geometry token (Eq. 1), which encodes area, equivalent diameter, circularity, and bounding-box ratios—i.e., the size implied purely by boundary morphology. A *3D protrusion estimate $s_{3D}$* is obtained from depth tokens using attention pooling, which aggregates local curvature, protrusion height, and coarse camera–lesion distance cues into a single scalar. Their difference $\Delta s = s_{2D} - s_{3D}$ quantifies the inconsistency between modalities; large values indicate cases where monocular appearance and pseudo-metric depth disagree. Empirically, $\Delta s$ correlates with prediction variance and serves as a surrogate indicator of epistemic uncertainty.

To calibrate the fused representation accordingly, SCB concatenates the multimodal embedding $z_{\text{fused}}$ with $(s_{2D}, s_{3D}, \Delta s)$ and feeds the result through a two-layer residual MLP:

$$z_{\text{cons}} = z_{\text{fused}} + f_{\text{SCB}}([z_{\text{fused}};\ s_{2D};\ s_{3D};\ \Delta s]). \tag{3}$$

This residual structure allows SCB to preserve $z_{\text{fused}}$ when cues are consistent, while learning corrective adjustments when 2D and 3D signals diverge. The calibrated representation $z_{\text{cons}}$ is finally mapped to a millimetre-level prediction via a regression head.

### 3.5. Prediction and Training Objective

The calibrated representation $z_{\text{cons}}$ is mapped to a millimetre-level estimate through a lightweight regression head,

$$\hat{y} = f_{\text{reg}}(z_{\text{cons}}).$$

A parallel auxiliary classifier predicts clinically relevant size bins (e.g., $< 5\,\text{mm}$, $5$–$9\,\text{mm}$, $\geq 10\,\text{mm}$), providing an ordinal signal that stabilizes learning near these thresholds. Although this multi-task formulation improves boundary sensitivity, the classifier serves purely as a training-time regularizer its outputs are discarded during inference, and MAE remains the primary evaluation metric. Classification metrics such as F1 are reported only to enable fair comparison with prior threshold-based systems. To obtain a single estimate per polyp, frame-level predictions across the video are aggregated using the $p75$ statistic, which emphasizes frames with clearer visibility and more reliable depth cues. This strategy reduces two inherent ambiguities of monocular endoscopy: systematic underestimation of large lesions and prediction instability around the $5\,\text{mm}$ diagnostic threshold.

## 4. Results and Discussion

**Dataset.**

We use the Polyp-Size dataset (Song et al., 2025), containing 42 polyps: 26 diminutive ($< 5\,\text{mm}$), 14 small ($5$–$10\,\text{mm}$), and only 2 large polyps ($> 10\,\text{mm}$), with per-polyp size annotations measured using calibrated vernier calipers. Each polyp appears in multiple

Table 1: Performance comparison between prior work, our baseline, and MPSE.

| Method | Accuracy | Recall | Precision | F1 | AUROC |
|---|---|---|---|---|---|
| Song et al (Song et al., 2025). | ∼0.65 | ∼0.65 | ∼0.67 | ∼0.61 | ∼0.69 |
| Our - Baseline | 0.738 | 0.938 | 0.600 | 0.732 | 0.933 |
| MPSE | 0.857 | 0.9375 | 0.800 | 0.857 | 0.945* |

consecutive frames, and all frames containing the polyp are used in training and evaluation. To prevent data leakage arising from strong temporal correlation across frames, all data splits are defined at the *polyp (video)* level rather than at the frame level. Specifically, frames belonging to the same polyp instance (i.e., extracted from the same endoscopic video segment) are assigned exclusively to either the training or the test split, but never both. This polyp-disjoint protocol ensures that the model is evaluated on entirely unseen lesions at test time, rather than benefiting from near-duplicate frames of the same polyp. Although multiple consecutive frames per polyp are used during training and evaluation, this setup reflects the intended clinical use case where size estimation is performed across short temporal windows of a single lesion.

### 4.1. Overall Performance

Table 1 and Figure 2 jointly illustrate the performance characteristics of MPSE relative to both the published benchmark and our internal baselines. The RGB and RGB–D models from Song et al. achieve moderate performance (accuracy ∼0.65, F1 ∼0.61) because depth is incorporated only through channel concatenation, without modelling geometric structure or scale ambiguity. Our RGB-only baseline improves these metrics (accuracy 0.738, F1 0.732) but Figure 2(a,c) reveals a systematic tendency to *underestimate* medium and large polyps, reflected in low precision (0.600) and inflated errors in the > 10,mm group. Adding depth alone increases recall substantially (0.938), yet precision remains low (0.600), and predic-

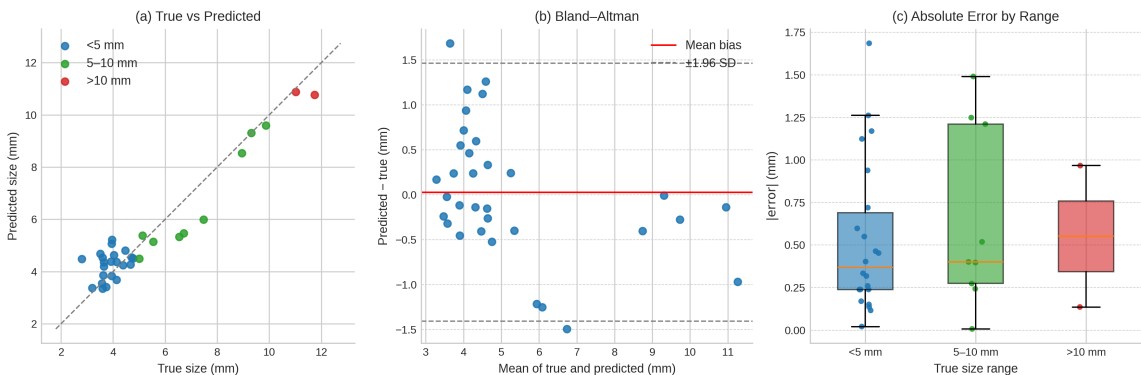

Figure 2: **MPSE Quantitative Evaluation.** **(a)** Predicted vs. true sizes show high correlation. **(b)** Bland–Altman plot shows near-zero bias (red line), confirming Scale Consistency Block efficacy. **(c)** Absolute error boxplots by size range

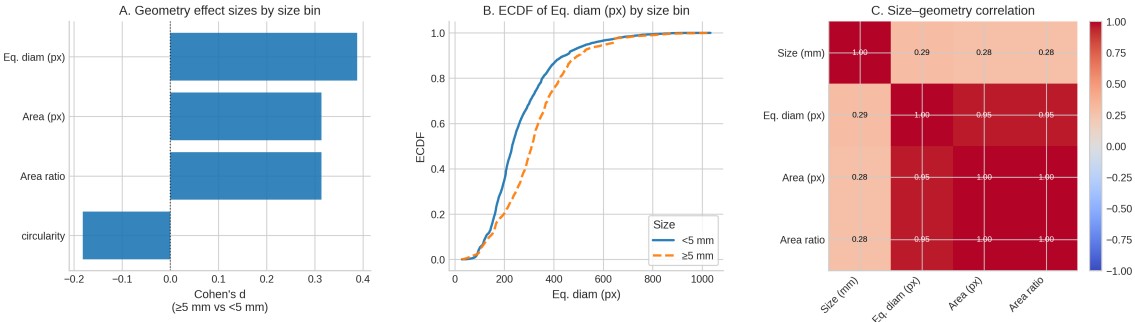

Figure 3: **Geometry feature analysis. (A)** Cohen's $d$ shows strong discriminative power for footprint cues (area, area ratio, equivalent diameter). **(B)** ECDFs **(C)** Correlation matrix between pixel-based metrics and true size

tions collapse toward the mid-size regime consistent with the Bland–Altman bias pattern in Figure 2(b). This behaviour is clinically concerning, as management decisions depend on accurate thresholds around 5,mm and 10,mm. MPSE, by contrast, integrates depth with explicit geometric priors and a scale-consistency mechanism, yielding balanced improvements across all metrics (accuracy 0.857, F1 0.857, AUROC 0.945*). The regression scatter shows tight alignment, the Bland–Altman plot shows near-zero bias, and the absolute-error boxplots confirm robustness even for large lesions (median $< 0.6$,mm for $> 10$,mm polyps). Together, these results demonstrate that MPSE does not simply perform better numerically it corrects the *structural failure modes* of prior RGB, RGB–D, and depth-only approaches by enforcing geometrically grounded, scale-aware multimodal reasoning.

## 4.2. Ablation Studies

### 4.2.1. Contribution of Geometry Features

Figure 3 shows that MPSE's geometry features are not auxiliary heuristics but encode structural information that is both discriminative and fundamentally incomplete. Panel A shows that simple footprint-based cues area, area ratio, and equivalent diameter—exhibit sizeable Cohen's $d$ between $< 5$ mm and $\geq 5$ mm lesions, indicating that geometry alone can reliably separate coarse size regimes. However, Panel B reveals substantial ECDF overlap, and Panel C confirms only weak correlation with true millimetre size ($r \approx 0.28$–$0.39$). Together, these plots expose a key property of monocular endoscopy: the 2D footprint is systematically confounded by camera–tissue distance and viewing angle, making geometry a stable but intrinsically underdetermined signal. MPSE therefore does not treat geometry as a stand-alone predictor but as a *structural prior* that constrains the fusion process. By elevating geometry to a guiding token, the model conditions how RGB and depth tokens are integrated, enabling detection of geometry–depth inconsistencies and preventing the mid-range collapse characteristic of appearance-only or naïve RGB–D approaches.

Table 2: **Critical component ablation on the validation set**

| ID | Variant | Depth | Geom | MPSE-Fuse | SCB | Best MAE ↓ |
|----|---------|-------|------|-----------|-----|------------|
| **B0** | **Baseline (MPSE)** | ✓ | ✓ | ✓ | ✓ | **0.927** |
| A1 | No Depth | | ✓ | ✓ | ✓ | **1.431** |
| A2 | No Geom | ✓ | | ✓ | ✓ | 1.354 |
| A3 | No MPSE-Fuse | ✓ | ✓ | | ✓ | 1.888 |
| A4 | No SCB | ✓ | ✓ | ✓ | | 1.231 |

### 4.2.2. CONTRIBUTION OF DEPTH FEATURES

While geometry captures "how big the lesion looks" in the image plane, depth is needed to answer "how big it really is" in three-dimensional space. Figure 4 shows this qualitatively: the RGB frames (top row) often present lesions whose apparent size is heavily influenced by camera distance and foreshortening, whereas the corresponding pseudo-depth maps (bottom row) reveal protrusion patterns and surface topography that are not obvious from intensity alone. The depth map $D$ is never used in isolation; instead, it is encoded into tokens that are jointly processed alongside RGB and geometry. In the Scale Consistency Block, depth contributes a coarse estimate of protrusion-based size ($s_{3D}$), which is contrasted with the footprint-based estimate ($s_{2D}$) derived from geometry. When these two disagree, MPSE learns to interpret whether the discrepancy indicates a genuinely protruded large lesion (e.g. high depth contrast, consistent surface) or unreliable depth (e.g. specular artefacts). The strong improvement in precision at matched recall in Table 1 is consistent with this mechanism: depth does not merely add noise, but resolves ambiguous 2D cases where geometry alone cannot disambiguate flat-close versus raised-far configurations.

### 4.2.3. EFFECTIVENESS OF THE CROSS-MODAL FUSION MECHANISM

The ablation study in Table 2 shows that each MPSE component contributes a distinct and non-redundant capability. Removing depth or geometry produces predictable failures: without depth, the model loses protrusion and distance cues; without geometry, it loses stable footprint statistics robust to illumination and motion. These behaviours align with the analytic trends in Figures 3. The Figure 4 highlights the complementary nature of geometry and pseudo-depth: while RGB frames provide the lesion footprint, the depth maps expose protrusion patterns and surface topography that resolve 2D ambiguity. Together, these cues explain why removing either modality (A1–A2) yields predictable collapse in accuracy. The severe degradation when MPSE-Fuse is removed (A3) demonstrates that

Table 3: **Robustness Evaluation of MPSE.**

| Condition | $\alpha$ | $\sigma_D$ | $\sigma_G$ / $p_{\mathrm{drop}}$ | **Polyp MAE (mm) ↓** |
|-----------|----------|------------|----------------------------------|----------------------|
| Baseline (clean) | 1.00 | 0.00 | 0.00 / 0.00 | 0.93 |
| Depth scale shift | 0.50 | 0.00 | 0.00 / 0.00 | 1.35 |
| Depth noise | 1.00 | 0.06 | 0.00 / 0.00 | 1.32 |
| Geometry corruption | 1.00 | 0.00 | 0.10 / 0.25 | 1.23 |
| Combined moderate failure | 0.75 | 0.03 | 0.05 / 0.10 | 1.39 |
| Combined severe failure | 0.50 | 0.06 | 0.10 / 0.25 | 1.48 |

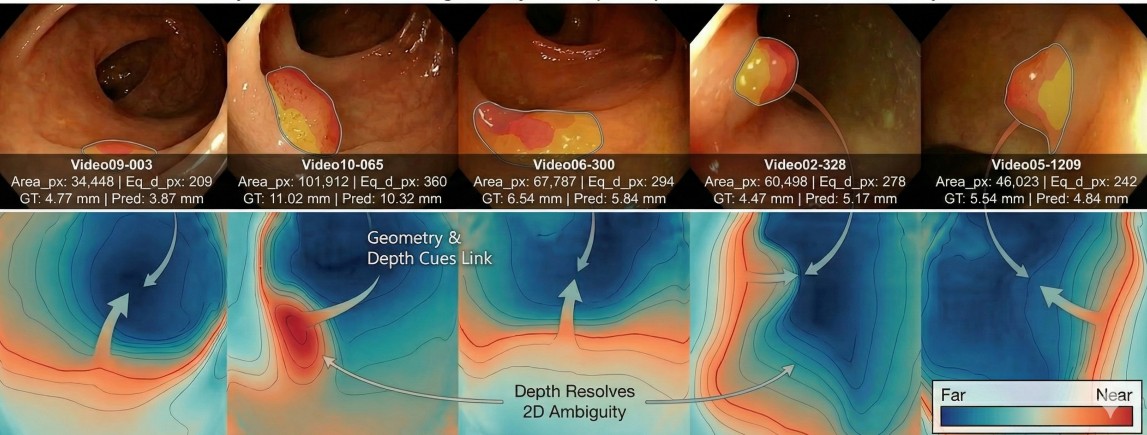

Figure 4: **Representative RGB frames and corresponding pseudo-depth maps.** The depth maps (bottom row) reveal 3D protrusion patterns and surface topography often missed by RGB appearance alone (top row).

performance arises not from stacking modalities, but from the *structured* geometry-as-query interaction that guides RGB and depth to informative regions. The Scale Consistency Block (A4) adds complementary robustness by reconciling disagreements between 2D and 3D cues. Its removal revives familiar biases underestimation of large lesions and instability near the 5 mm threshold highlighting its role in resolving cases where naive fusion fails. Overall, the ablations paint a coherent picture: geometry and depth are individually necessary, directed fusion is indispensable, and SCB corrects the remaining scale ambiguity inherent to monocular endoscopy.

Table 3 evaluates the robustness of MPSE under controlled degradations of depth and geometry cues at inference time, without retraining, to emulate deployment-relevant failure modes in clinical endoscopy such as depth scale ambiguity, sensor noise, and imperfect segmentation. Perturbations are injected directly into the depth maps and geometry feature representations, with geometry corruption applied at the feature level rather than to raw binary masks, since ground-truth segmentation masks are not available for this dataset and segmentation quality in practice varies across cases. This design allows us to assess sensitivity to generalized geometric uncertainty independent of the specific artifact patterns of any single upstream segmentation model. Across all tested conditions, including severe combined perturbations, polyp-level MAE remains bounded and increases gradually relative to the clean baseline, with no evidence of abrupt performance collapse. Notably, worst-case errors are comparable to ablations that remove individual components (Table 2), indicating that MPSE does not rely critically on any single modality or precise metric calibration. Overall, these results demonstrate controlled and predictable degradation under adverse conditions, supporting the robustness of MPSE in noisy, real-world endoscopic deployment.

### 4.2.4. Limitations observed

Despite its strong performance, MPSE exhibits two predictable failure modes arising from intrinsic limits of monocular endoscopy rather than architectural shortcomings. First,

polyps larger than 10 mm are occasionally underestimated due to scarce supervision in the right tail, the nonlinear scaling of footprint geometry with camera distance, and depth-map saturation on smooth near-field surfaces—precisely the regime where monocular 3D cues become unreliable. Multiple interventions produced only modest gains, confirming that this bias is structurally rooted rather than an artifact of model capacity or training instability. Accordingly, results in the $> 10$ mm regime should be interpreted as indicative of systematic bias correction relative to RGB and naïve RGB–D baselines, rather than statistically representative performance across large lesions. Second, ambiguity persists around the clinically important 5 mm threshold: lesions in the 4–6 mm range often exhibit nearly identical footprints and shallow depth gradients, making sub-millimetre discrimination inherently difficult under monocular viewing conditions.

## 4.3. Clinical Implications

Reliable estimation of polyp size is central to risk stratification, resection planning, and surveillance interval determination. In particular, thresholds at 5 mm and 10 mm guide the adoption of resect-and-discard and diagnose-and-leave strategies, yet visual estimation remains highly variable even among expert endoscopists. By integrating depth-derived structural cues with segmentation-based geometric priors, our tri-modal framework reduces the scale ambiguity that commonly leads to underestimation of protruded or irregular lesions. The improved recall observed near the clinically sensitive 5 mm boundary suggests that MPSE can serve as a stabilizing decision-support tool in real-time workflows, especially in community settings where advanced measurement devices such as VSE systems are unavailable.

## 5. Conclusion and Future Work

We presented MPSE, a geometry-aware, depth-guided multimodal fusion framework for reliable polyp size estimation from monocular endoscopy. By jointly leveraging RGB appearance, segmentation-derived geometry descriptors, and pseudo-metric depth cues, the model overcomes key limitations of traditional image-based measurement, including scale ambiguity and inconsistent predictions near the 5 mm clinical threshold. Future work will explore several directions. First, incorporating specialized endoscopy depth models or self-supervised geometric pretraining may further enhance 3D reasoning under specular or low-texture conditions. Second, integrating temporal transformers could provide more robust cross-frame aggregation for long sequences and mitigate transient segmentation noise. Third, expanding evaluation to multi-center datasets and diverse imaging conditions will be essential for assessing generalizability. Finally, coupling size estimation with uncertainty quantification and downstream tasks such as resection recommendation offers a promising route toward clinically comprehensive decision-support systems.

## Appendix A. Appendix

### A.1. Implementation Details

Unless otherwise stated, RGB and pseudo-depth inputs were resized to the training resolution used by the respective backbone encoders. The RGB and depth streams used ResNet-18 backbones. Geometry descriptors included pixel area, equivalent diameter, aspect-ratio features, circularity, and a boundary-sharpness measure derived from the segmentation mask. The fused representation was processed with a lightweight transformer encoder with $L = 2$ layers, followed by the Scale Consistency Block and a regression head for millimetre-level prediction.

### A.2. Data Split Protocol

All experiments were performed using polyp-disjoint splits. Frames from the same polyp instance were assigned exclusively to either training or test partitions to prevent leakage from near-duplicate temporal frames. Frame-level predictions were aggregated at the polyp level using the $p75$ statistic.

### A.3. Additional Note on Limitations

Performance on lesions larger than $10\,\text{mm}$ should be interpreted with caution due to the very limited number of large polyps in the dataset. This limitation is inherent to the available dataset and motivates broader future evaluation on larger multi-center cohorts.

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
