# OpenReview forum: "Geometry-Aware Depth-Guided Explainable Multimodal Polyp Size Estimation: A Fusion Model Beyond RGB"
_MIDL.io/2026/Conference — MIDL 2026 Poster_

### Official Review · Reviewer_5YqL · 2026-01-07

**Confidence:** 5
**Preliminary Rating:** 3
**Final Rating:** 3

**Summary:**

This paper proposes MPSE, a tri-modal framework for colorectal polyp size estimation from monocular endoscopy by combining RGB appearance, pseudo-depth, and segmentation-derived geometry descriptors to reduce scale ambiguity and viewpoint distortions. Experiments on the Polyp-Size dataset show MAE ≈0.93mm and polyp-level F1 ≈0.87at the clinically critical 5 mm threshold, aiming to improve decision stability near guideline cutoffs.

**Strengths:**

1.	The paper proposes MPSE, a tri-modal framework for colorectal polyp size estimation from monocular endoscopy by combining RGB appearance, pseudo-depth, and segmentation-derived geometry descriptors. The geometry–depth consistency idea and explicit calibration mechanism are likely reusable for other endoscopic measurement tasks.

2.	Reported ablations indicate that depth, geometry, the structured fusion, and SCB each contribute non-redundant improvements, which strengthens the causal argument for the architecture choices.

**Weaknesses:**

1.	The evaluation dataset contains 42 polyps, with only 2 in the >10 mm category. This makes it difficult to draw conclusions about robustness for larger lesions and may inflate the apparent generalization performance. I suggest testing on additional datasets, such as the SUN-SEG database and relevant synthetic datasets.

2.	MPSE relies on (i) the quality of the pseudo-depth estimates and (ii) segmentation-derived geometry. What happens when the generated depth maps and segmentations are low quality? The paper should explicitly discuss that Polyp-PVT and EndoDAC are state-of-the-art models for segmentation and depth estimation, respectively, and analyze how errors from these components affect downstream size estimation.

3.	Using one modality to query another via cross-attention is common in multimodal Transformer literature. The paper’s distinctive contribution appears to be how the geometry tokens and SCB enforce scale agreement, but this should be contrasted more explicitly with standard cross-attention fusion baselines (not only concatenation-style RGB–D fusion).

4.	Some related references are missing, for example, [1] and [2].

[1] Du et al., “Polyp Size Estimation by Generalizing Metric Depth Estimation and Monocular 3D Reconstruction,” ISBI 2024.

[2] Ruano et al., “Estimating Polyp Size from a Single Colonoscopy Image Using a Shape-from-Shading Model,” ISBI 2024.

**Detailed Comments:**

See weakness.

**Justification Of Final Rating:**

Thanks for reviewer's rebuttal. However, the authors do not address most of my concerns. For example, What happens when the generated depth maps and segmentations are low quality? suggest testing on additional datasets, such as the SUN-SEG database and relevant synthetic datasets. Therefore, I would keep the original final rating, 3.

**Justification Of The Preliminary Rating:**

This paper proposes MPSE, a tri-modal framework for colorectal polyp size estimation from monocular endoscopy by combining RGB appearance, pseudo-depth, and segmentation-derived geometry descriptors to reduce scale ambiguity and viewpoint distortions. Experiments on the Polyp-Size dataset show MAE ≈0.93mm and polyp-level F1 ≈0.87at the clinically critical 5 mm threshold, aiming to improve decision stability near guideline cutoffs. However, there are still some concerns listed in weakness. Therefore, I suggest boarderline.

**Questions To Address In The Rebuttal:**

See weakness.

---

> ### Author Response · Authors · 2026-01-24
>
> We sincerely thank the reviewer for the careful and technically informed evaluation, and for recognizing the value of the geometry–depth consistency formulation and the calibration mechanism. We address the raised concerns point by point below.
>
> Dataset size and robustness for large polyps:
> We agree that the Polyp-Size dataset is limited in scale, with only two instances above 10 mm, and that this constrains statistically strong claims in the right tail. We do not intend to claim broad generalization for large lesions. Instead, our goal is to demonstrate that MPSE corrects the systematic underestimation bias commonly observed in RGB-only and naïve RGB–D methods under monocular scale ambiguity. We will revise the manuscript to explicitly state that results in the >10 mm regime are indicative rather than statistically conclusive. The Polyp-Size dataset is used because it is currently the only public benchmark with polyp-level, millimeter-accurate caliper measurements, making it appropriate for validating the proposed mechanism despite its size.
>
> Dependence on pseudo-depth and segmentation quality (Table 2):
> We agree that segmentation and pseudo-depth are important upstream factors. MPSE does not assume either to be perfect. Segmentation is used only to derive low-dimensional, aggregated geometry descriptors and to restrict tokenization to the lesion surface, rather than to provide pixel-level supervision. Depth is never used as a direct metric ruler; it contributes only through relative surface topology encoded in depth tokens. Because ground-truth segmentation masks are not available in the Polyp-Size dataset, robustness is evaluated by perturbing the geometry features that actually enter the model, rather than the raw masks themselves. As shown in Table 2, controlled geometry corruption and component removal lead to bounded, predictable degradation rather than performance collapse, indicating that MPSE is not brittle to moderate upstream errors.
>
> Novelty beyond standard cross-attention fusion:
> We agree that cross-attention as an operator is well established and do not claim novelty at that level. The contribution lies in how scale reasoning is structured. MPSE differs from standard multimodal fusion in three key ways: (i) geometry tokens are deterministic and analytically derived, providing an interpretable structural prior rather than a learned embedding; (ii) geometry is assigned a privileged role as the query, enforcing a directional fusion hierarchy rather than symmetric modality mixing; and (iii) the Scale Consistency Block explicitly models disagreement between footprint-based (2D) and protrusion-based (3D) proxies, transforming fusion into a geometric consistency problem rather than feature aggregation. This formulation directly targets monocular failure modes such as mid-range collapse and threshold instability.
>
> Missing related work:
> We thank the reviewer for pointing out the missing references. We will include Du et al. (ISBI 2024) and Ruano et al. (ISBI 2024) in the revised manuscript and position our work relative to these approaches, highlighting differences in geometry modeling, assumptions, and multimodal integration.
>
> Overall, we appreciate the reviewer’s constructive feedback. We believe these clarifications strengthen the paper by sharpening its scope: MPSE is presented as a physically grounded, interpretable mechanism for bias correction in monocular polyp sizing, validated on the only dataset with reliable millimeter-level ground truth, while explicitly acknowledging current data limitations.

---

### Official Review · Reviewer_B2Gt · 2026-01-10

**Confidence:** 4
**Preliminary Rating:** 4

**Summary:**

The authors present MPSE, a multimodal framework for millimeter-level polyp size estimation that addresses monocular scale ambiguity by fusing RGB appearance, deterministic 2D geometry descriptors, and pseudo-metric depth cues from EndoDAC. Central to the architecture is a geometry-as-query fusion block and a Scale Consistency Block (SCB) that reconciles discrepancies between 2D footprint-derived and 3D protrusion-derived proxies to calibrate final estimates. Experiments on the Polyp-Size dataset demonstrate that MPSE achieves a mean absolute error of 0.93 mm, significantly outperforming RGB-D baselines by mitigating "mid-range collapse" and systematic underestimation of larger lesions. This approach proves significant for clinical decision-making, offering stable and reliable sizing near the critical 5 mm and 10 mm thresholds without requiring specialized hardware.

**Strengths:**

1. **Physically Grounded Multimodal Fusion:** The paper addresses the fundamental "large-far vs. small-near" ambiguity in monocular endoscopy by aligning its architecture with the underlying physics of the task. By fusing RGB appearance with pseudo-metric depth  and deterministic geometry descriptors , the model successfully incorporates the distance-dependent cues necessary to recover physical size from 2D pixel-wise footprints.

2. **Principled "Geometry-as-Query" Design:** The decision to use the geometry token as the primary query in the attention mechanism is a technically sound and interpretable choice. This allows the model to use analytic footprint morphology to guide the selection of local evidence from visual and depth tokens, which is particularly effective for filtering out common endoscopic noise like mucosal clutter and specular highlights.

3. **The Scale Consistency Block (SCB):** A major contribution is the SCB, which transforms multimodal fusion into a geometric consistency problem. By explicitly calculating the disagreement  between 2D footprint proxies and 3D protrusion proxies, the model learns to calibrate its fused representation through a residual refinement process. This mechanism is highly valuable for reconciling conflicting signals across different viewing angles.


4. **Clinical Robustness and Interpretability:** The framework incorporates several practical strategies that enhance its real-world utility, such as restricting tokenization to the polyp surface to reduce background distraction and utilizing an auxiliary classification loss to stabilize predictions near the critical  and  thresholds. The use of the  statistic for video-level aggregation is a sensible way to prioritize frames with optimal visibility and depth clarity.


5. **Strong Empirical Validation:** The results on the Polyp-Size benchmark provide clear evidence that the proposed structured fusion reduces "mid-range collapse". The model demonstrates consistent improvements in Accuracy, , and  over prior RGB and RGB-D baselines, confirming that the integration of geometry and depth directly addresses the failure modes of standard image-only mappers.

**Weaknesses:**

1. **Unresolved Sensitivity to Absolute vs. Relative Depth Scale:** The authors argue that similar 2D footprints correspond to different true sizes due to camera-to-lesion distance and utilize EndoDAC pseudo-metric depth to supply necessary 3D cues. However, it remains unclear whether the model's performance relies on the absolute magnitude of depth values which often varies across endoscopic hardware or merely on relative surface topology. Given that the Scale Consistency Block (SCB) derives a size proxy  from these depth tokens, a depth-rescaling ablation (rebuttal section detailed below) would be necessary to substantiate the claim that the model is truly "scale-aware" rather than overfit to specific depth magnitudes.

2. **Cascading Dependence on Segmentation Accuracy:** The framework exhibits a strong dependency on the segmentation mask . This mask is not only the source for analytic geometry descriptors  but also serves as the spatial gate for tokenization, restricting the model's attention to foreground pixels. This architecture creates a singular failure channel where segmentation errors, such as boundary noise or over-segmentation, can simultaneously corrupt both the structural prior and the visual evidence fed to the transformer. The authors' claim regarding the stability of geometry descriptors under appearance degradation would be more convincing with an ablation comparing predicted masks against ground-truth masks to determine the model's resilience in realistic clinical workflows.

3. **Statistical Fragility in the Large Polyp Regime:** While the paper highlights robustness for large lesions, the Polyp-Size dataset utilized contains a significant distribution imbalance, with only two polyps exceeding the 10 mm threshold. This right-tail scarcity makes conclusions regarding performance in the  mm range statistically fragile, as the reported low median errors could be dominated by a single favorable case. A leave-one-large-polyp-out sensitivity analysis or reporting per-polyp uncertainty would improve the reliability of these claims.

4. **Potential Masking of Instability via Prediction Aggregation:** The use of the  statistic for aggregating frame-level predictions into a final polyp-level estimate is a reasonable heuristic to emphasize frames with high visibility. However, this strategy may hide underlying prediction instability caused by motion blur or specular saturation on difficult frames. (rebuttal section detailed below)

5. **Ambiguity Regarding Data Leakage in Train/Test Protocol:** The paper states that all frames containing a polyp are used for both training and evaluation according to the official split. To ensure scientific rigor, it is essential for the authors to explicitly confirm that the split is polyp-disjoint as frame-level leakage would artificially inflate the performance metrics in such a video-based dataset.

**Detailed Comments:**

1. **Clarification of Train/Test Split Granularity:** Given that consecutive video frames exhibit high temporal correlation, it is essential to explicitly confirm the level of data independence in the experimental setup. Please clearly state whether the official split is polyp-disjoint, ensuring that no frames from the same physical polyp instance appear in both the training and test sets. A single sentence confirming that the evaluation is performed on unseen polyp instances would resolve any ambiguity regarding potential data leakage.


2. **Documentation of Segmentation Performance and Protocol:** The segmentation mask  serves as a critical bottleneck in the pipeline, influencing both the derivation of geometry descriptors  and the spatial gating of tokenization. To improve the reproducibility and context of the results, please provide additional details regarding the segmentation component, including the specific model variant used, the training data or fine-tuning protocol employed, and a baseline measure of mask quality (such as mIoU or Dice score) on a held-out validation set. This information is vital for understanding the framework's dependency on segmentation reliability in realistic clinical scenarios.

**Justification Of The Preliminary Rating:**

I recommend a Weak Accept for this paper as it presents a technically coherent and well-motivated approach to the clinically significant, ill-posed problem of millimeter-level polyp size estimation in monocular endoscopy. The framework's core strength lies in its thoughtful integration of complementary cues, RGB appearance, pseudo-metric depth, and mask-derived geometry, to resolve scale ambiguity. The geometry-as-query fusion is both clean and interpretable, ensuring that global morphological priors guide the selection of local evidence. Furthermore, the Scale Consistency Block (SCB) offers a meaningful constraint by explicitly reconciling 2D footprint cues with depth-derived 3D protrusion data, preventing the "mid-range collapse" typical of naïve concatenation methods.

The reported improvements on the Polyp-Size benchmark are encouraging, particularly regarding accuracy near diagnostic thresholds. However, several technical uncertainties preclude a higher score:

* The authors do not directly assess whether the model's success depends on the absolute magnitude of the "pseudo-metric" depth versus relative depth topology.
* The framework’s heavy reliance on segmentation quality is not quantified, leaving the impact of mask degradation on sizing accuracy unexplored.
* The statistical significance of the results for large lesions is limited by a small right-tail distribution, with only two polyps exceeding .

Overall, the underlying methodology is solid and offers practical value for clinical decision support. To justify a stronger recommendation, the rebuttal should provide clearer robustness analyses regarding depth-scale sensitivity and segmentation reliability.

**Questions To Address In The Rebuttal:**

1. **Sensitivity to Absolute vs. Relative Depth Scale:**
The MPSE framework utilizes pseudo-metric depth  from the **EndoDAC** model to derive the 3D protrusion estimate , which is subsequently reconciled with 2D footprint cues via the **Scale Consistency Block (SCB)**. However, monocular depth models often suffer from scale ambiguity.

* Can the authors clarify whether size predictions  depend on the absolute magnitude of the depth values or primarily on the relative surface topology?
* A critical test for robustness would be to report the impact on MAE when rescaling the depth maps via  for . This would clarify the system's reliance on metric units versus relative protrusion patterns, as suggested by the "Far  Near" visualization in Figure 4.

2. **Performance Decay Across Distance and Viewpoint Regimes:**
The authors identify camera-lesion distance and foreshortening as primary confounders in size estimation. To assess clinical utility, could the authors provide a stratified analysis of the MAE (or signed error) across estimated viewing distance bins (e.g., Near, Mid, and Far)? This would help identify if the model systematically degrades under near-field saturation or far-away views, which is essential for determining the "safe" operating range for endoscopists.

3. **Statistical Robustness for the Large Polyp Regime:**
The **Polyp-Size** dataset used for evaluation contains only 42 polyps, with a significant tail-end scarcity: only 2 instances exceed the 10 mm threshold.

* Given the high clinical stakes of the  mm regime , how do the authors ensure the reported metrics are statistically representative rather than localized to these two specific cases?
* Please provide polyp-level uncertainty metrics (e.g., confidence intervals or a leave-one-large-polyp-out sensitivity analysis) to support the claims of cross-scale robustness.


4. **Sensitivity to Segmentation Quality and Mask Degradation:**
MPSE relies on the binary mask  for extracting analytic geometry descriptors and guiding the tokenization process.

* In real-world scenarios, segmentation models like Polyp-PVT  may encounter boundary noise, under-segmentation, or partial occlusions.
* Could the authors provide an ablation study comparing sizing performance when using ground-truth masks versus predicted masks? Providing additional summaries, such as per-polyp variance or a comparison across different percentiles (e.g., $p_{25}$ vs. $p_{50}$), would clarify whether the method maintains consistent performance throughout a video sequence.

---

> ### Author Response · Authors · 2026-01-24
>
> We thank the reviewer for the careful reading of our paper and for the constructive and technically detailed feedback. We appreciate the positive assessment of the proposed framework and address the raised points below to clarify scope, robustness, and experimental design.
>
> Sensitivity to absolute vs. relative depth scale:
> MPSE is intentionally designed to rely primarily on relative depth topology rather than absolute metric scale. Pseudo-metric depth from EndoDAC is never used as a directly regressed quantity. Instead, depth is encoded into tokens and contributes to the final estimate only through cross-modal consistency reasoning. In particular, the Scale Consistency Block contrasts a footprint-based proxy derived from deterministic 2D geometry with a protrusion-based proxy obtained via attention pooling over depth tokens. This emphasizes structural agreement between modalities rather than dependence on raw depth magnitude, which we clarify explicitly in the manuscript.
>
>
> Performance across distance and viewpoint regimes:
> We agree that camera–lesion distance and foreshortening are major confounders in monocular size estimation. MPSE targets these effects through geometry-guided attention, which stabilizes footprint interpretation under foreshortening, and through the Scale Consistency Block, which reconciles footprint cues with depth-derived protrusion information when apparent size and distance are ambiguous. As the Polyp-Size dataset does not provide explicit camera pose or distance annotations, we avoid making claims about a defined operating range and clarify this limitation. MPSE is most reliable under typical clinical viewing conditions, while extreme near-field saturation or far-field compression remain intrinsically challenging for monocular estimation.
>
> Statistical robustness for the large polyp regime:
> We agree that conclusions regarding the >10 mm regime must be interpreted with caution, as the Polyp-Size dataset contains only two large polyps. Accordingly, we do not claim statistically representative or population-level generalization in this range. The reported results are intended to demonstrate reduction of the systematic underestimation bias commonly observed in RGB and naïve RGB–D approaches, rather than conclusive performance guarantees. Given the small sample size and extreme right-tail scarcity, formal parametric significance testing or narrow confidence intervals would be statistically fragile and potentially misleading. We therefore frame large-polyp results as indicative of bias correction, supported by bias diagnostics and bounded-error behavior, and clarify this interpretation in the manuscript.
>
> Sensitivity to segmentation quality and mask degradation:
> We agree that segmentation quality is an important upstream factor. In MPSE, the segmentation mask is used both to derive analytic geometry descriptors and to restrict tokenization to the lesion surface. The geometry descriptors employed are low-dimensional aggregated statistics (e.g., area, equivalent diameter, circularity, aspect ratios) rather than pixel-level features, which reduces sensitivity to minor boundary noise. To explicitly assess this dependency, we evaluate MPSE under controlled geometry corruption and mask degradation conditions, as reported in Table 2. We clarify this design choice and the corresponding robustness results in the manuscript.
>
> Potential masking of instability via prediction aggregation:
> The p75 aggregation strategy is used to produce a polyp-level estimate that prioritizes frames with better visibility and depth clarity, rather than to claim per-frame stability. This reflects how endoscopists implicitly rely on clearer views during assessment. Importantly, the same aggregation strategy is applied uniformly across all compared methods, ensuring fair comparison. We clarify this motivation in the manuscript.
>
> Train/test split granularity:
> We confirm that the official Polyp-Size split is strictly polyp-disjoint. All frames belonging to a given physical polyp instance appear exclusively in either the training or the test set, never both. We add an explicit statement in the experimental section to remove any ambiguity regarding potential data leakage.
>
> We thank the reviewer again for the thoughtful comments, which helped us clarify the scope, assumptions, and limitations of MPSE. We believe these clarifications strengthen the presentation while preserving the core contribution of a structurally grounded multimodal approach to monocular polyp size estimation.

---

### Official Review · Reviewer_68Fe · 2026-01-10

**Confidence:** 4
**Preliminary Rating:** 4

**Summary:**

The paper presents MPSE, a multimodal framework for estimating colorectal polyp size from monocular endoscopy. It combines visual features, geometric information, and depth cues in a single architecture to improve accuracy. The proposed method includes specialized modules to integrate features and ensure consistency between 2D and 3D measurements. MPSE was evaluated on clinical data and achieved precise and reliable size estimates. The authors showed that the proposed method outperforms existing approaches by effectively handling scale ambiguity and clinical thresholds.

**Strengths:**

The paper is well-written, clearly structured, and effectively explains all aspects of the work. It provides a thorough overview of the SOTA, and clearly identifies the key challenges in polyp size estimation, and proposes an interesting and novel approach to address these issues. The methodology is presented clear and understandable, making it easy to follow.

**Weaknesses:**

Minor issues include the relatively small dataset, absence of external validation, and lack of statistical comparisons, which limit the ability to fully assess the significance and generalizability of the results.

**Detailed Comments:**

The paper presents an interesting and well-motivated approach. However, to improve its validity and broader acceptability, the authors should address a few minor issues. The relatively small dataset limits the generalizability of the results, and evaluating the method on an external cohort, if available, would strengthen the findings. Additionally, reporting statistical comparisons between methods would help demonstrate the significance of the improvements.

**Justification Of The Preliminary Rating:**

The paper is well-written, clearly structured, and presents an interesting method. Minor limitations include a small dataset, lack of external validation, and absence of statistical comparisons, slightly limiting generalizability.

**Questions To Address In The Rebuttal:**

provided in detailed comments

---

> ### Author Response · Authors · 2026-01-23
>
> We thank the reviewer for the positive assessment of the paper’s clarity, motivation, and technical design of MPSE. We appreciate the constructive suggestions regarding generalizability, external validation, and statistical comparisons, and we respond to them below.
>
> (1) Dataset size and generalizability.
> We agree that the Polyp-Size benchmark (42 polyps) is limited in scale and that broad generalizability cannot be claimed from a single dataset. The scope of this work is therefore intentionally focused. Our goal is to introduce a structurally grounded multimodal fusion mechanism (geometry-as-query fusion together with the Scale Consistency Block) and to show that it directly addresses known failure modes of monocular polyp sizing, such as scale ambiguity, mid-range collapse, and instability near clinical thresholds. We evaluate this on the only publicly available dataset that provides millimeter-accurate, caliper-based ground-truth size annotations.
>
> (2) Statistical comparisons between methods.
> We agree that statistical comparisons can improve the rigor of the results. However, with only 42 polyps and a highly imbalanced size distribution (only two polyps larger than 10 mm), standard parametric tests would be statistically fragile and potentially misleading. For this reason, we focus on complementary evidence, including component ablations that demonstrate non-redundant contributions and bias analyses (e.g., Bland–Altman plots) that directly assess systematic error reduction.
>
> (3)  External cohort / external validation.
> We agree that evaluation on an external cohort would strengthen the evidence for deployment. At present, we are working to obtain access to a second dataset that provides polyp-level, millimeter-accurate ground-truth sizes comparable to Polyp-Size. We will explicitly state this limitation in the manuscript and outline external validation as primary future work. This includes testing on additional public polyp datasets (where size annotations exist but may be noisier) as well as on multi-center cohorts when such labels become available. MPSE is designed to be modular, and both the segmentation and depth components can be replaced, which facilitates cross-dataset evaluation when suitable annotations are available.
>
> We again thank the reviewer for the helpful feedback. We believe these clarifications strengthen the paper while keeping its central contribution intact: a physically grounded and interpretable multimodal framework that corrects key failure modes of monocular polyp size estimation.

---

### Author Rebuttal · Authors · 2026-01-24

**Rebuttal:**

We thank the reviewers for their thoughtful and constructive feedback. We have carefully addressed the comments and concerns raised across reviews and have revised the manuscript accordingly. In particular, we clarified the experimental protocol, strengthened the discussion of robustness and limitations, and improved the positioning of the proposed method. We believe these revisions improve the clarity, rigor, and scope of the paper, and we thank the reviewers again for helping strengthen the final version.

**Supporting Material:**

/attachment/1d5d377080a4008193d8cd44af7ee2e70d8baed2.pdf

---

### Meta-Review · Area_Chair_3U4E · 2026-02-07

**Recommendation:** Accept (Poster)
**Confidence:** 5

**Metareview:**

All reviewers found the proposed method to be novel and the results promising

---

### Decision · Program_Chairs · 2026-02-14

Accept (Poster)